# Impact of integrating family planning with maternal and child health on uptake of contraception: A quasi-experimental study in rural, Sindh, Pakistan

Zahid Memon[1]*, Wardah Ahmed[1], Shah Muhammad[2], Fatima Haider[3], Talib Hussain Lashari[4], Muhammad Jawwad[2], Sophie Reale[5], Rachael Spencer[5], Zulfiqar Bhutta[2], Hora Soltani[5]

1 Department of Community Health Sciences, Aga Khan University, Karachi, Sindh, Pakistan, 2 Center of Excellence in Women and Child Health, Aga Khan University, Karachi, Sindh, Pakistan, 3 Department of Family Medicine, Aga Khan University, Karachi, Sindh, Pakistan, 4 Population Welfare Department, Government of Sindh, Karachi, Sindh,Pakistan, 5 Sheffield Hallam University, Sheffield, United Kingdom

* zahid.memon@aku.edu

## Abstract

This study aimed to evaluate the impact of integrating family planning with maternal and child health (FP-MCH) on the uptake of modern contraceptive methods (MCMs) and related health outcomes in two rural districts of Sindh, Pakistan using a quasi-experimental control before-after study design. Intervention package integrated FP with MCH in the existing public sector at facility and community levels. This included capacity building of health care providers, ensuring sustained supplies of family planning commodities, and community engagement activities through Lady Health Workers (LHW). Data was collected through household surveys at baseline (1st October- 31st December 2020) and endline (1st October- 31st December 2022). Sample size was calculated as 880 married women of reproductive age (MWRA) in each district. The Difference-in-Differences (DiD) analytical method was used to assess the impact of intervention. There was an increase of 11.7% in current use of MCMs in the intervention group compared to the control group (p-value <0.003), with increases observed in uptake of injections, implants, and condoms. Furthermore, family planning counseling during ANC (DiD 9.1%, p-value 0.162), LHW visits during pregnancy (DiD 15.4%, p-value 0.018), postnatal care (PNC) visits for mothers (DiD 24.0%, p-value <0.001), LHW visits after delivery (DiD 20.6%, p-value <0.001), and counseling by LHW about family planning at PNC visit (DiD 15.3%, p-value 0.027). The study concludes that integrating FP with MCH services proved impactful in increasing contraception uptake and minimizing missed opportunities. It underscores the necessity for cohesive efforts by the government and local stakeholders to design local, regional, and national policy frameworks pertaining to health and population planning for sustainable mother and child health improvements.

**Data availability statement:** The data will be made available with reasonable request to Imran Ahmed (imran.ahmed@aku.edu), Director-Data Management Unit, Centre of Excellence in Women and Child Health, Aga Khan University. The data file and code book uploaded as Supporting Information file. 2 and 3.

**Funding:** "This research was funded by Global Affairs Canada through United Nations Population Fund (UNFPA), grant number SMK532400. The funders had no role in study design, data collection and analysis, decision to publish, or preparation of the manuscript."

**Competing interests:** The authors have declared that no competing interests exist.

## Introduction

Pakistan made slow progress towards improving maternal and child health (MCH) indicators within South Asia [1,2]. Pakistan's contraceptive prevalence rate (CPR) has not progressed beyond 35% since 2007 and unmet need stands at 21% [1], despite the upscaling of family planning (FP) programs, dedicated budget allocations and political will drive by commitments to FP 2030 and the Sustainable Development Goals (SDGs). The local literature reveals that these efforts are substantially hampered by limited accessibility and availability of FP services, specifically in rural areas, lack of healthcare provider skills and capacity in FP, contraceptive commodity insecurity, socio-cultural stigma and barriers, poor community level knowledge, education and promotion of FP and birth spacing practices and at the individual level, lack of women's agency and autonomy in decisions relating to FP and their sexual and reproductive health and rights [3–7].

The significance of family planning (FP) in Pakistan's commitment to maternal and child health is emphasized [8,9]. Fulfilling unmet needs through FP is reported to potentially reduce maternal deaths by 30% [1,10]. Additionally, contraceptive-induced birth spacing may contribute to a 10% decrease in under-5 child mortality [11,12]. However, to date, the country has failed to embed integrated services within the scope of its maternal and child health continuum of care, leading to missed opportunities for contraceptive access particularly modern contraceptive methods, service delivery, and uptake [13]. In keeping with this evidence, the province of Sindh has attempted to resolve this issue by identifying strategic areas such as supply-side service delivery mechanisms and addressing the community and household-level barriers. These areas have been outlined in the Sindh province's Costed Implementation Plans (CIP), developed as part of the FP 2030 commitments [14] and commitment to SDG.

Recognizing this gap, a project Sihaat Mand Khaandan (SMK)-building healthy families, was undertaken in the province of Sindh in 2020 with the collaboration of department of health (DoH). The study involved designing and implementing a culturally appropriate integrated Family Planning and Maternal Child Health (FP-MCH) service deliver model as an intervention in rural districts with the broader aim of improving access to healthcare services, addressing health inequities and disparities related to family planning, and aligning FP-MCH services with government priorities and goals.

Initially a systematic review of South Asian countries was conducted [15]. The review underscores the importance of integrating demand-generation interventions into family planning (FP) programs, particularly in South Asian regions. These interventions, targeting individual and community-level barriers and socio-cultural norms, show significant potential in increasing modern contraceptive usage and reducing unmet FP needs to achieve reproductive goals. While interventions within the existing healthcare system or new franchising options effectively enhance accessibility, the disparity in FP uptake between urban and rural areas in South Asia calls attention to geographical inequities. Targeted efforts can create a conducive environment for intervention implementation, ultimately improving reproductive, maternal,

and child health outcomes and mitigating urban-rural disparities [16]. Later, using a qualitative exploratory design was employed among community member, men, and women, to explore beliefs on modern contraceptives [17]. Interviews with healthcare workers delved into intersections between FP and reproductive health service delivery at facility and outreach levels. The study highlights significant barriers to FP uptake, including women constrained financial autonomy, limited mobility, and cultural norms affecting their decision-making in rural Sindh. Facility-level challenges such as stock-outs of contraceptives and inadequate healthcare capacity also deter women from seeking services. The lack of integration between family planning and maternal and child health services is identified as a missed opportunity. Moreover, demand-side barriers like spousal disapproval, social stigma, and concerns about contraceptive side effects further hinder family planning uptake.

The project implemented the FP -MNCH integration model to assess its influence on the uptake of modern contraceptives in rural Sindh. Therefore, the primary objective of this study is to evaluate how the FP-MNCH integration model affects the uptake and coverage of FP services in the targeted population.

The purpose of conducting this evaluation is to strengthen the existing base of local evidence on the FP-MCH integration model, particularly its impact measurable as the increase in FP uptake in the context of rural Sindh. Such evidence can serve as a source of crucial input to the design of the local, regional, and national policy frameworks pertaining to health and population planning. More importantly, it can provide important recommendations on the way forward for strengthening health systems through multi-sectoral coordination and collaboration aimed at empowering women, mothers, and young girls in all spheres of life.

## Materials and methods

### Study design

This study used Quasi-Experimental (QE) with control before-after study design. Baseline and end-line cross sectional surveys were conducted in the intervention and control districts to assess the impact of implementing an integrated FP-MCH delivery model on the uptake of MCMs. A complex integrated intervention was implemented in the intervention district, and routine services continued in the control district for the period between July 2021-September 2022 (15 months). Full details about the study design, setting, and methodology can be accessed in the published protocol paper [18]. The baseline survey was implemented at the population level on 1st October- 31st December 2020, followed by a follow-up household survey on 1st October- 31st December 2022.

### District selection

The study was implemented in two districts of the province of Sindh in Pakistan, Matiari and Badin-(Fig 1).

The interventions were rolled out within existing health facilities and their catchment outreach LHWs. The district of Badin was selected as a control district using the Propensity Score Matching (PSM) approach taking household as the unit of analysis. PSM is a statistical method to create a propensity score for each participant in the intervention group based on the essential characteristics or covariates. Several key variables used for PSM calculation including Unmet Need (UMN), Contraceptive Prevalence Rate (CPR), Modern Contraceptive Prevalence Rate (mCPR), Human Development Index (HDI), Female literacy and Maternal Mortality (FIMM), Deliveries by Skilled Birth Attendants (DSBA), and Antenatal Care (ANC) coverage. By this matching Matiari and Badi were identified as the closest match, with a propensity score of 0.31 for Matiari and 0.36 for Badin. Further details on the matching process are provided in the supporting information file (S1 Table). These scores matched with the control group at baseline, thus reducing the problem of comparison across multiple variables [19]. In this study, matching was completed using a sampling frame comprising 23 rural districts of Sindh province using the Multiple Indicator Cluster Survey (MICS) 2018 [20].

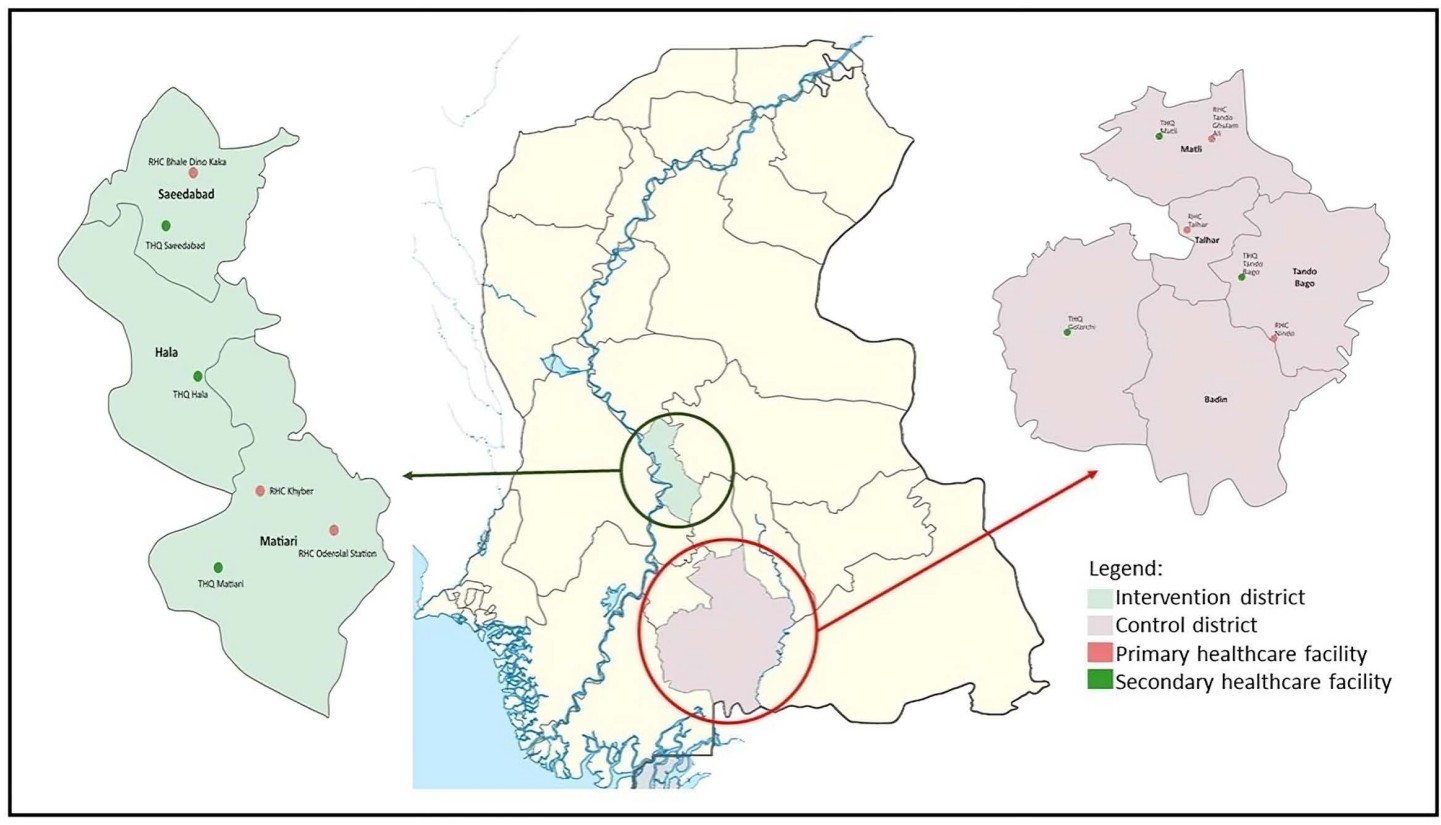

**Fig 1. Location\* of the primary and secondary health care facilities in the intervention (Matiari) and control (Badin) district. RHC (Rural Health Center)-primary level care facility; THQ- secondary level care facility.\*** Map developed on ArcGIS, available here for proper citation of base map: https://support.esri.com/en-us/knowledge-base/faq-what-is-the-correct-way-to-cite-an-arcgis-online-ba.000012040#:~:text=To%20reference%20the%20use%20of,Scale. Compatible with our CC-BY 4.0 license.

## District Matiari: Intervention

District Matiari is 185 km from the provincial capital Karachi. It has a population of about 770,040, with the following characteristics. The majority of the population, 76.2%, lives in rural areas, 42.6% are literate: 54.05% for males and 38.48% for females, and 65% are below 30 years of age. Administratively, Matiari is divided into three sub-districts: Hala, Saeedabad, and Matiari, with 30 Union Councils and 112 villages.

Regarding health systems, Matiari has 20 Basic Health Units, 4 Rural Health Centers, 2 Tehsil Headquarter Hospitals, 1 District Headquarter Hospital, and 14 dispensaries with a network comprising 429 LHWs, supervised by 17 Lady Health Supervisors.

## District Badin: Control

District Badin, Sindh, is located 212 km from Karachi city. It has a population of about 1.8 million, of which 78.4% are rural, 33.7% are literate, and around 70% are below 30 years of age. Badin is divided into five sub-districts: Shaheed Fazil Rahu, Badin, Talhar, Tando Bago, and Matli; 46 Union Councils; and 535 villages. The health system of Badin comprises 45 Basic Health Units, 11 Rural Health Centers, 5 Tehsil Headquarter Hospitals, 1 District Headquarter Hospital, and 66 dispensaries with a network of about 1,100 LHWs, supervised by 36 Lady Health Supervisors.

## Study sites

Six public health facilities and all LHWs working in their catchment areas were included in the intervention within Matiari. Purposively selected facilities and aligned their scope of services provision and patient flow based on the catchment population ratio. Later, the health facilities, including their catchment population in the intervention district, were selected and finalized in consultation with various stakeholders (mainly government officials). Overall, six primary and secondary level facilities, such as 3 Rural Health Centers (RHCs) and 3 Taluka Head Quarters (THQ) in the intervention district-Matiari, were selected with same level 6 facilities in the control district Badin.

## Study intervention

FP-MCH integrated model-(Fig 2) was comprised of a set of interventions, which were implemented at the health facility and community level to increase access and use of FP [21].

The existing staff along with the project team implemented it in close collaboration with and oversight from DoH and Population Welfare Department (PWD), Government of Sindh. Detailed comparison of intervention package between intervention and control district is tabulated in Table 1.

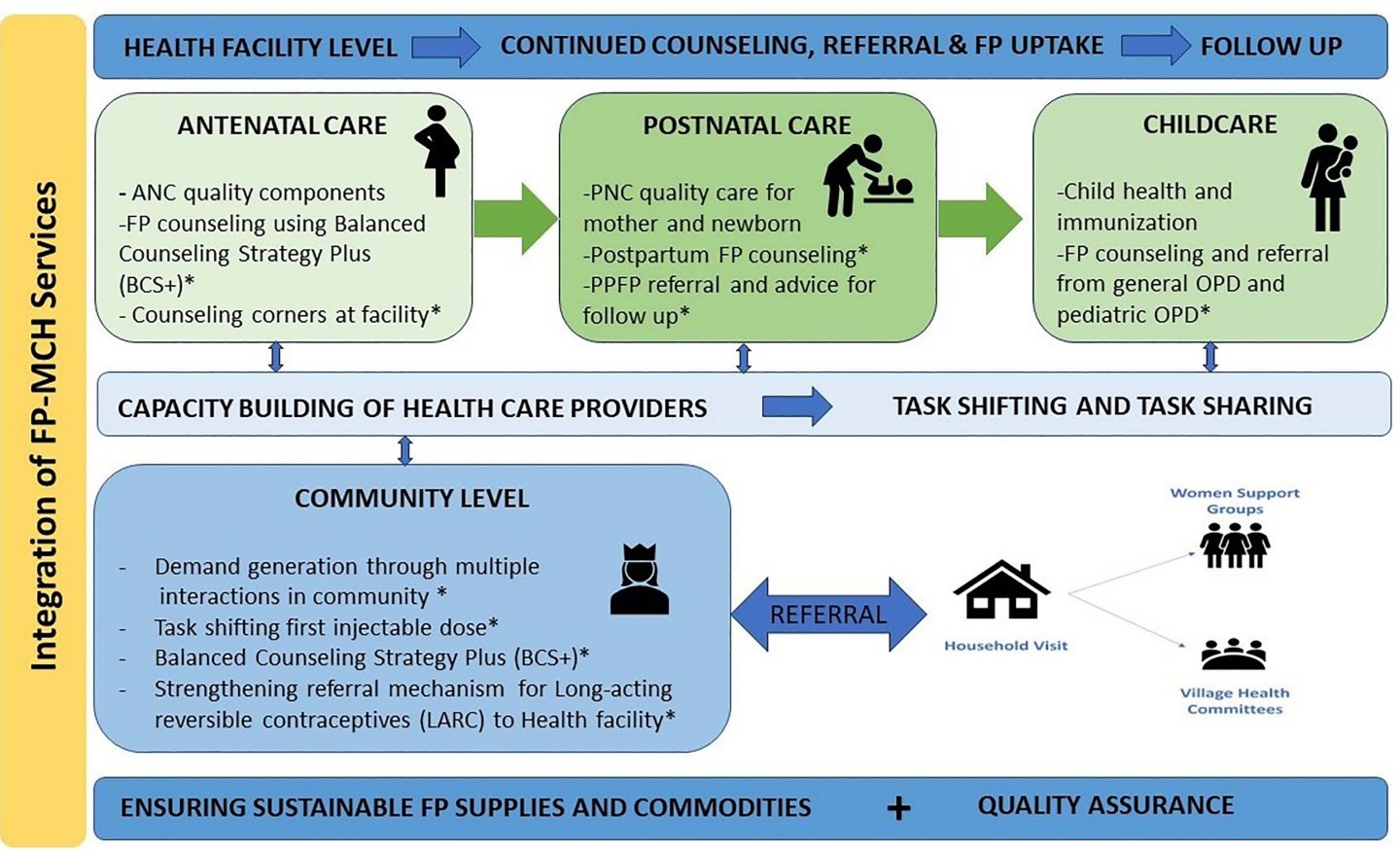

**Fig 2. Integrated FP-MCH (Family planning – maternal and child health) services model.** *Contact points for integration of FP and MCH services in addition to routine services.

**Table 1. Description of integrated family planning and maternal and child health (FP-MCH) intervention package in intervention district as compared to control district (standard activities).**

| Integrated interventions | Interventions district | Control district |
|---|---|---|
| **Health Facility Level** | | |
| FP counseling corners established at health facilities in OPD areas to provide counseling to pregnant women on postpartum family planning. | √ | X |
| Placement of FP and MCH services related Information, education, and communication (IEC) material at facility and LHW-health houses. | √ | X |
| Inter and intra facility referrals for FP from ANC, PNC, childcare OPDS and general OPDs and counseling. | | |
| **Capacity building activities** | | |
| Training of health care providers on Balanced Counseling Strategy Plus (BCS+) including Women Medical Officers, Pediatrician, Nurses, Community Midwives and Lady Health Visitors and Lady health Workers. | √ | X |
| Training EPI vaccinators on referring individuals for family planning counseling and services at healthcare facilities (HCF). | √ | X |
| **Task Shifting and task Sharing**: Training and certification of Nurses, Community Midwives and Lady Health Visitors on LARC and Lady health Workers on the provision of first dose of injectable contraceptives. | √ | X |
| **Community level** | | |
| Project staff reviews LHW monthly reports and awareness sessions | √ | X |
| LHW household visits in their catchment population | √ | √ |
| LHW household visits + identification of eligible women for contraception and referral to HCF | √ | X |
| Use of Balanced Counseling Strategy Plus (BCS+) to promote FP services uptake | √ | X |
| Regular Women Support group (WSG) sessions | √ | √ |
| Regular Village Health Committees (VHC) sessions | √ | √ |
| Revitalize WSG sessions focusing on FP and MCH at their health houses | √ | X |
| Revitalize Village Health Committees for males focusing FP and MCH | √ | X |
| **Supplies of FP Commodities at Facility and Community levels** | | |
| Supplies provided by government through routine supply chain mechanism | √ | √ |
| One day training on Contraceptive Logistic Management Information System (cLMS/CLR6) | √ | X |
| Training on stock management – timely requisition and follow up | √ | X |

The integrated intervention package included the following:

1  Integration of FP-At three points at the health facility level-Antenatal care (ANC), Postnatal Care (PNC), and pediatric services including OPD and immunization counter. At the level of ANC separate counseling corners were established for FP counseling provided by trained healthcare providers. At the level of PNC, postpartum FP (PPFP) counseling and service provision was ensured. Furthermore, the pediatric HCPs and EPI vaccinators were trained to counsel and refer the potential clients to avail FP services.

2  Capacity Building-Training of facility-based health care providers (HCP) and community health workers on counseling skills, technical skills, task shifting and task sharing to expand service provision. Training on strengthening FP counseling skills was based on Balanced Counseling Strategy Plus (BCS+) (third edition) and job aids were distributed [22,23]. Training and certification were conducted to enable the provision of the first dose of subcutaneous injectable contraceptives to LHWs and to enable the provision of Long-Acting Reversible Contraceptives (LARC) to mid-level healthcare providers at the facility level.

3  Integration at the community level-Revitalizing women support group (WSG), village health committee (VHC) sessions and made their health houses functional to inform the community regarding FP choices, promote FP referrals from community, and help in decision-making by under the supportive supervision of LHS.

4   Strengthening supply chain of FP commodities-District health office (DHO) personnel, administrative staff from health facilities were trained on Contraceptive Logistic Management Information Systems (cLMS/CLR6). Participants were trained in FP stock management, including calculating demands for FP commodities, data entry and reporting, and FP stock requisition. (Results of these interventions are beyond the scope of this manuscript).

## Study methods

### Sampling technique

The two-stage sampling technique selected an eligible study population in both surveys. The targeted population for this study consisted of MWRA between 15–49 years of age residing in the study districts within the catchment population of public sector health facilities. The respondents for knowledge and practices of MCH services were married women of reproductive age with at least one child under 2 years of age at the time of data collection in both sites. In the first stage, 44 clusters were randomly selected from each study district's list of clusters. A cluster was defined as a catchment population of 1000–1500 served by an LHW linked to the selected health facilities. In the second stage, households with eligible women (MWRA) were identified by conducting a line listing of the eligible households. A total of 20 households per cluster were selected randomly from the list of eligible households.

### Sample size calculation

The sample size for the study was calculated based on the prevalence of the primary outcome indicator: the use of modern contraceptive methods. A total of 880 Married Women of Reproductive Age 15-49 years (MWRA) were required in each group (district) for each round of the survey. This sample size was sufficient to detect an increase from 28.9% to 36.9% (i.e., an 8% increase) in the proportion of MCM uptake with 95% CI and 80% power [20]. The sample size calculations accounted for the assumed design effect of 1.5 and a 7% nonresponse rate. For analysis, 860 and 824 respondents at baseline and 880 and 873 at endline were interviewed in the intervention and control districts, respectively.

### Data collection procedure

The data was collected on all relevant program indicators, including contraceptive prevalence rate, use of ante-post-natal services, skilled birth attendance, and socio-demographic characteristics. Data collectors were trained on face-to-face interview-based structured questionnaire adapted from Pakistan Demographic Health Survey (PDHS).

The study was conducted in each district by an assigned data collection team comprising one team leader and four data collectors. Before data collection commenced, line listing was performed in selected clusters. Data was collected using personal digital assistants (Model Samsung SM T289) through an Android application designed for field-based data collection, with each session lasting 45–60 minutes. All data was uploaded electronically, analyzed, and displayed on the virtual project dashboard. To maintain confidentiality, coded data was stored in a password-protected and encrypted system, accessible only to authorized personnel. Privacy was ensured throughout the interview, which was conducted in a separate room or space within the household with the index respondent.

### Data management and analysis

The collected data were analyzed in STATA version 17 (Stata Corp, Texas), and bivariate statistical tests and independent sample t-test were used to summarize the data. Sampling weight was applied by using a survey setting. Frequency and percentage were computed for categorical variables and mean, and standard deviation (SD) were calculated for continuous variables.

Descriptive statistics were used to show the demographic characteristics, including the household size that's the number of household members living in the house for more than three months. Respondent's age in years, number of living children, respondent's age and education, and her husband's education categorize as no education, primary, middle, secondary and intermediate or above. Additionally, the wealth quantile was used as an indicator of the distribution of wealth in the population covered. A wealth index was constructed using principal components analysis and predicted using the first principal component. The index was calculated at the household level and based on the description in demographic health survey (DHS) survey procedures [24]. The score was categorized into five equal quintiles, namely poorest, poorer, middle, richer, and richest, with the first quintile representing the poorest 20% and the fifth quintile representing the richest 20% of the population.

Chi-square was used to show associations between basic demographic characteristics and study outcome (increased MCM and improved MCH services). Difference-in-differences (DiD) analysis used to evaluate the program or policy made a difference by comparing changes over time between two groups (intervention and control). First, assess how things were before the program started then how things changed for both groups over time. Finally, compare the changes. If the intervention group changed more, the program might have a real impact. It helps rule out other factors that could have influenced the outcomes. The differences of household characteristics at baseline in the intervention and control areas were controlled from linear mixed models with a log link using binomial distribution for categorical variables and Gaussian for continuous variables, and cluster as a random effect.

We compared changes from baseline to endline in the two districts using a DiD analysis. Unadjusted and multivariable DiD estimates were obtained from mixed linear regression models with an interaction term between variables for districts (intervention vs. control) and time (endline vs. baseline) and cluster as a random effect. We applied DiD for change in the prevalence of MCMs and the use of each contraceptive method as primary outcome indicators. Also, we assessed changes in secondary outcomes indicators: women who sought antenatal care, four or more antenatal care visits and postnatal care visits (within 48 hours of home and facility-based delivery) at health facility level, facility birth, skilled birth attendant, received advice for family planning methods from Lady Health Workers (as part at their routine visit in during pregnancy and postnatal care visit) and fully immunized child (0–23months, with BCG vaccine, three doses of polio & pentavalent vaccines and 1 dose of measles vaccine). The primary and secondary outcomes adjusted for respondent's age, respondent's education, wealth quantile, household size (family members), and number of pregnancies for uptake of MCM. The limited data set and codebook files are available as supporting information file (S1 Data; S1 Codebook).

### Ethical approval

Ethical approval was granted by the Ethical Review Committee (ERC) of Aga Khan University on July 16, 2020 (2020-3606-18261). The study protocol was also approved by the National Bioethics Committee of Pakistan (4–87/NBC-514/20/231). Verbal informed consent was obtained from married women (respondents), with the risks and benefits of participation explained in the presence of an impartial witness and guardian if the respondent was under 18 years of age. Respondents were given the opportunity to ask questions about the study before providing consent, except for those who declined to participate. Given that the population-based household survey posed minimal risk and maintained anonymity using unique identifiers, verbal consent was considered sufficient [25]. Trained data collectors read aloud the simplified informed consent and assent in the local language (Sindhi) to ensure comprehension. The data collectors recorded the verbal informed consent and assent by noting the "agreement to participate" in the presence of the head of the household, a witness, and a guardian who was at least 18 years old. The consent process was supervised by the team leader.

## Results

### Socio-demographic characteristics of the study respondents in control and intervention districts

The Table 2 showed the average/mean household size was 7.5 ± 2.2 in the intervention district, whereas 7.0 ± 3.3 in the control district at baseline. The mean age of the respondent was 33 years, parity was 4, and most women

PLOS Global Public Health

**Table 2. Socio-demographic characteristics of the study respondents (community) in Intervention and Control districts (Baseline).**

| Variable Baseline | Intervention (Matiari) N = 860 | | Control (Badin) N = 824 | | Absolute Difference | p value* (t-test/ Chi-square) |
|---|---|---|---|---|---|---|
| | mean | SD | mean | SD | | |
| Household Size | 7.5 | (±2.2) | 7.0 | (±3.3) | 0.5 | 0.006 |
| Respondent's age | 33.4 | (± 7.8) | 31.3 | (± 6.5) | 2.2 | <0.001 |
| Husband's age | 38.1 | (± 9.6) | 34.8 | (±7.5) | 3.3 | <0.001 |
| Number of Children (Parity) | 4.0 | (±2.0) | 3.5 | (± 2.0) | 0.4 | <0.001 |
| Number of Pregnancies | 4.5 | (±2.4) | 4.0 | (± 2.3) | 0.5 | 0.006 |
| | n | % | n | % | % | |
| Mother's Education | | | | | | |
| No education | 619 | (73.4) | 612 | (74.2) | (-0.8) | 0.891 |
| Primary | 121 | (14.2) | 103 | (12.9) | (1.3) | 0.645 |
| Middle | 25 | (2.8) | 33 | (4.1) | (-1.3) | 0.252 |
| Secondary | 42 | (4.3) | 26 | (3.5) | (0.8) | 0.654 |
| Intermediate or above | 53 | (5.3) | 50 | (5.3) | (0.0) | 0.984 |
| Wealth Quantile | | | | | | |
| Poorest | 59 | (6.6) | 276 | (32.9) | (-26.3) | <0.001* |
| Poor | 167 | (20.2) | 169 | (20.2) | (-0.1) | 0.990 |
| Middle | 214 | (25.4) | 125 | (16.0) | (9.4) | 0.009* |
| Rich | 202 | (24.0) | 136 | (16.6) | (7.4) | 0.039* |
| Richest | 218 | (23.8) | 118 | (14.2) | (9.4) | 0.090 |

*p value less than 0.05 was considered as statistically significant.

73.4% were uneducated in the intervention district. In terms of socioeconomic status, 6.6% of the respondents in the intervention group and 32.9% in the control group belonged to the poorest wealth quantile with a p-value < 0.001.The significant baseline differences between intervention and control districts were adjusted for in the DID model.

### Effects of intervention on primary outcomes

In the present study, socio-demographic characteristics, including household family size, mother's age, husband's age, and number of pregnancies, were adjusted in the DiD analysis. Table 3 provides DiD results on the knowledge and use of MCMs comparing baseline with a follow-up an endline survey. The DiD estimate showed a statistically significant increase of 10.3% p-value of 0.012 in the current use of any method to delay pregnancy in the intervention group compared to the control group. The current use of MCMs shows an 11.7% increase in the intervention group compared to the control group with a p-value of 0.003. Overall, intervention packages showed a noticeable impact in FP uptake.

Moreover, Table 3 shows the prevalence of various MCMs among the study respondents at baseline and endline. There was a significant increase in the uptake of injections, and condoms in the intervention group. The DiD estimates show an 8.7% percentage point positive difference with p-value less than 0.001, indicating a significant increase in the use of the injection in the intervention sites as compared to the control group. Similarly, condom uptake increased by 5.6% with p-value of 0.003. The use of the implant method showed a positive increase of 1.3%. However, the p-value of 0.291 was not significant. Further, the analysis demonstrated a 1.0% reduction in traditional contraceptive methods at the endline in the intervention group.

**Table 3. Difference-in-Differences analysis for current contraceptive use by method.**

| Indicators | Baseline survey | | | | | | Endline survey | | | | | | DiD¥ adjusted | |
|---|---|---|---|---|---|---|---|---|---|---|---|---|---|---|
| | Intervention (Matiari) N = 860 | | Control (Badin) N = 824 | | Absolute difference | | Intervention (Matiari) N = 880 | | Control (Badin) N = 873 | | Absolute difference | | Net effect diff¥ | p value* |
| | n | % | n | % | % | | n | % | n | % | % | | % | |
| Currently using any method to delay pregnancy | 242 | (28.0) | 220 | (26.7) | (1.3) | | 353 | (40.0) | 251 | (28.4) | (11.6) | | (10.3) | 0.012 |
| Currently using a modern method to delay pregnancy | 224 | (25.9) | 217 | (26.4) | (-0.5) | | 334 | (37.9) | 224 | (26.9) | (11.0) | | (11.7) | 0.003 |
| Female Sterilization | 68 | (8.0) | 41 | (5.2) | (2.8) | | 62 | (7.3) | 52 | (5.3) | (2.0) | | (-1.4) | 0.552 |
| Male Sterilization | 0 | (0.0) | 1 | (0.2) | (-0.2) | | 0 | (0.0) | 0 | (0.0) | (0.1) | | (0.2) | 0.312 |
| IUD** | 7 | (0.9) | 5 | (0.5) | (0.4) | | 10 | (1.1) | 3 | (0.2) | (0.8) | | (0.6) | 0.339 |
| Injection | 43 | (5.6) | 70 | (8.6) | (-3.0) | | 85 | (9.5) | 26 | (4.5) | (5.0) | | (8.7) | <0.001 |
| Implants | 26 | (2.9) | 10 | (1.2) | (1.6) | | 39 | (4.3) | 11 | (1.7) | (2.7) | | (1.3) | 0.291 |
| Pills | 34 | (3.9) | 47 | (5.5) | (-1.6) | | 62 | (7.0) | 51 | (7.6) | (-0.6) | | (0.5) | 0.888 |
| Condom | 45 | (4.7) | 42 | (5.1) | (0.4) | | 66 | (7.5) | 28 | (2.4) | (5.2) | | (5.6) | 0.003 |
| Standard Days Method (SDM) | 1 | (0.1) | 0 | (0.0) | (0.1) | | 1 | (0.1) | 4 | (0.3) | (-0.2) | | (-0.4) | 0.183 |
| Lactation Amenorrhea Method (LAM) | 0 | (0.2) | 1 | (0.1) | (-0.1) | | 9 | (1.1) | 49 | (5.0) | (-3.9) | | (-3.4) | 0.003 |
| Currently using a traditional method to delay pregnancy | 18 | (2.1) | 6 | (0.7) | (1.4) | | 19 | (2.1) | 27 | (1.6) | (0.5) | | (-1.0) | 0.384 |
| Rhythm method | 1 | (0.1) | 0 | (0.0) | (0.1) | | 0 | (0.0) | 2 | (0.2) | (-0.2) | | (0.2) | 0.102 |
| Withdrawal | 17 | (2.0) | 4 | (0.5) | (1.5) | | 19 | (2.1) | 25 | (1.4) | (0.7) | | (0.7) | 0.372 |

*p value less than 0.05 was considered as statistically significant.

**Intra Uterine Device (IUD).

¥Adjusted for cluster, Mother age, Mother Education, Wealth Quantile, Household size (Family members), No of pregnancies.

## Effects of intervention on secondary outcomes

The DiD estimates of secondary outcomes are shown in Table 4. Information collected on ANC indicators proportion of respondents "sought ANC", "4 more ANC visits", "family planning counseling during ANC visits", "LHW visit during pregnancy", "facility births", "skilled birth attendant", "postnatal visit of mother", "LHW visit after delivery", "LHW counseling for family planning as a part of PNC visit", "Newborns checkup after birth" and "fully vaccinated children" from 0-23 months as child health indicator.

For FP counseling during ANC visits, the results show that at baseline, 2.0% in the intervention area received counseling compared to 7.9% of respondents in the control area. At the endline, the proportion increased to 37.5% in the intervention and 30.5% in the control area. DiD estimates revealed an increase of 9.1% as the intervention impact, although the p-value was not significant 0.162.

Regarding LHW visits during pregnancy, 87.0% were reported in control districts at baseline, increasing to 98.7% at the endline. A similar trend was observed in the intervention district; at baseline, 55.1% of respondents were visited by LHW, which increased to 82.5% at the endline. DiD estimates showed a significant and positive increase of 15.4% with p-value of 0.018.

Further, adjusted DiD estimates show statistically significant improvements in facility births (DiD 15.4%, p-value <0.001), skilled birth attendants (DiD 16.0%, p-value <0.001), PNC visits for mothers (DiD 24.0%, p-value <0.001), LHW visit after delivery (DiD 20.6%, p-value <0.001), LHW counseling for family planning in PNC visit (DiD 15.3%, p-value

**Table 4. Difference-in-differences analysis for secondary outcomes.**

| Indicators | Baseline survey | | | | | | Endline survey | | | | | | DiD¥ adjusted | |
|---|---|---|---|---|---|---|---|---|---|---|---|---|---|---|
| | Intervention (Matiari) N=413 | | Control (Badin) N=448 | | Absolute difference | | Intervention (Matiari) N=395 | | Control (Badin) N=427 | | Absolute difference | | Net effect diff¥ | p value* |
| | n | % | n | % | | | n | % | n | % | | | | |
| ANC Sought | 379 | (92.8) | 387 | (86.4) | (6.3) | | 378 | (95.9) | 396 | (94.4) | (1.5) | | (-5.0) | 0.090 |
| 4 or more ANC Visits | 205 | (49.1) | 166 | (37.9) | (11.2) | | 228 | (57.7) | 235 | (47.6) | (10.1) | | (-1.4) | 0.819 |
| FP counseling during ANC Visits | 11 | (2.0) | 34 | (7.9) | (-5.9) | | 149 | (37.5) | 139 | (30.5) | (7.0) | | (9.1) | <0.162 |
| LHW Visited during pregnancy | 214 | (55.1) | 389 | (87.0) | (-31.9) | | 323 | (82.5) | 420 | (98.7) | (-16.2) | | (15.4) | 0.018 |
| Facility births | 304 | (75.3) | 374 | (83.7) | (-8.4) | | 374 | (94.6) | 388 | (86.7) | (7.9) | | (15.0) | <0.001 |
| Skilled Birth Attendant | 305 | (75.5) | 373 | (83.5) | (-8.0) | | 377 | (95.4) | 388 | (86.3) | (9.1) | | (16.0) | <0.001 |
| PNC checkup-mother | 180 | (43.9) | 337 | (75.4) | (-31.6) | | 275 | (70.2) | 344 | (78.1) | (-7.9) | | (24.0) | <0.001 |
| LHW Visited after delivery | 155 | (38.9) | 363 | (81.3) | (-42.5) | | 299 | (75.8) | 415 | (97.5) | (-21.7) | | (20.6) | <0.001 |
| LHW counseling for family planning in PNC visit | 72 | (19.5) | 118 | (25.4) | (-5.9) | | 81 | (20.5) | 182 | (40.4) | (-19.9) | | (-15.3) | 0.027 |
| Newborns check up after birth | 245 | (60.7) | 363 | (80.8) | (-20.1) | | 303 | (77.3) | 397 | (90.5) | (-13.2) | | (5.5) | 0.312 |
| Fully Immunized Child[a] | 108 | (37.7) | 72 | (21.9) | (15.7) | | 234 | (59.4) | 269 | (64.3) | (-4.9) | | (-17.0) | 0.027 |

*p value less than 0.05 was considered as statistically significant.

¥Adjusted for cluster, Mother age, Mother Education, Wealth Quantile, Household size (Family members), No of pregnancies.

[a]For children 0–23. Age-appropriate vaccination: BCG, three doses of DPT-HEPB-HIB, four doses of oral polio vaccine, one dose of inactivated polio vaccine, three doses of pneumococcal vaccine, and one dose of measles vaccine.

0.027) and fully immunized children (DiD -17%, p-value 0.027). Moreover, although there was increase in newborn checkup after birth DiD 5.5% however the p-value was not significant 0.312.

## Discussion

This study aimed to evaluate the impact of integrating family planning with maternal and child health (FP-MCH) on the uptake of modern contraceptive methods (MCMs) and related health outcomes in rural intervention district-Matiari as compared to control district-Badin in the province of Sindh, Pakistan.

The findings revealed that implementing an integrated model for FP-MCH service delivery through the existing public health facilities and LHW program has impacted the uptake of MCMs within 15 months of implementation. The intervention district showed an increase in the contraception uptake, particularly the injections, and condoms.

Although the results from a quasi-experimental study should be interpreted cautiously. The increased uptake of FP and MCH services suggests that the program is acceptable with the intended targeted beneficiaries. This is an encouraging result that shows the success of the integrated model at both the facility and community level. The results reflected through the difference-in-differences (DiD) analysis demonstrated an increase in the utilization of antenatal care services, skilled birth attendants, postnatal checkups, and LHWs visits as an important finding of the integrated service delivery model.

The existing literature reveals various definitions of integration. Whilst some consider integration of services as a point of care, others consider it at the level of national programs as a mechanism for strengthening service delivery through monitoring and evaluation, integrated supply chains and technical support [26]. Within the context of this research, FP-MCH integration applies to the former with a focus on points of care in the existing health system such as antenatal care, postnatal care, delivery services as well as child immunization services. This is the first time in district Matiari that comprehensive and multiple contact points were involved in providing integrated FP-MCH health services including

counseling at health facilities and in the community. The aim was achieved to reinforce the delivery of FP messages at each contact point. Integration involves utilizing FP and MCH services through various entry points, promoting structural coherence, achieving efficiencies, and adopting holistic approaches for individuals with diverse health requirements [27,28]. The comprehensive package of services effectively addressed the interrelated health needs of women and children across the continuum of care. It emphasized the importance of offering a more coordinated approach to health care and service delivery [29,30]. A similar study conducted in Sri Lanka on the integration of FP services within MCH settings reported a 15% increase in uptake of modern contraception amongst women, in comparison to those accessing non-integrated services [31].

From the findings of this study, it appears FP-MCH integrated is a double-edged model, that in turn also improves MCH indicators. An improvement was noted in the attendance of the ANC visits, which are the first point of contact of a pregnant woman with trained healthcare providers for receiving quality ANC care services. Aligning with the results, it was also demonstrated by a study in Nepal in which integrated FP services were implemented within MCH clinics and resulted in enhanced antenatal care attendance [32]. Furthermore, a similar study conducted in clinical settings in Nepal reported a 30% increase in adoption of postpartum contraception in integrated clinics compared to those that were non-integrated. This was reflected by a 24% increase in the number of women attending postnatal care visits [33]. Consistent with our findings, other studies conducted in similar settings in Niger, Burkina Faso, and India have demonstrated that focusing on pregnant women and mothers proven as effective platforms for identifying individuals who may benefit from FP methods and referring them for counseling within health facilities [34]. Similarly, ANC coupled with FP counseling is more likely to increase the facility births through skilled birth attendants and subsequent postnatal checkups for mothers and newborns [35].

Despite evidence of the success of integration, there are inherent challenges with its implementation [36,37]. A study in India found that integration propelled disparities in access to FP-MCH services, with women residing in rural areas reporting 20% lower access to integrated services in comparison to their urban counterparts [38]. Our study addressed this issue by implementing task shifting and task sharing as one of the interventions in the FP-MCH integration model, which expanded the overall coverage of FP services to hard-to-reach areas along with shifting the workload from the healthcare providers at the facility level [39].

The health care providers in LMICs perceive they are not sufficiently trained to provide integrated FP services specifically with regard to identifying which contraceptive methods are safe for clients to use (based on their personal medical history and any related complications) and skills for counseling on various FP methods [40]. In addition to it, the most reported reason for not opting for healthcare services at public facilities was the poor attitude of doctors and nurses [41]. Considering these gaps, capacity building was kept as an integral part of this study integration model where the health workforce was trained on balanced counseling strategy BCS+ along with other technical trainings. Literature shows empathetic counseling improves acceptability and informed decision-making for uptake of method of choice [42,43].

Integrating FP with other maternal and child health services results in fewer missed opportunities [44]. Counseling and referrals from ANC, PNC, pediatric OPD, general OPD and immunization clinics by trained health care providers was recognized as a promising approach for improving FP uptake. Moreover, encouragement for postpartum FP (PPFP) counseling during PNC visits resulted in greater access to relevant FP services facilitating healthy birth spacing. Integration allows for improving service delivery and is acceptable for clients, providers, and community members in limited resource settings [28,45,46]. This integration and the complex interventions was further supported by synthesizing evidence from High Impact Practice (HIPS) and following strategies that could facilitate successful integration of family planning and child health services during the postpartum period, leveraging existing healthcare platforms [47].

In addition to the health facilities, the integration model was also adapted at the community level through LHWs. Trained LHWs tailored their sessions according to communities and individual needs and preferences [48]. This fostered

group discussions around MCH topics, encouraging social support and understanding of the importance of FP uptake at their women support groups (WSG) and village health committees (VHC) sessions [49]. Based on earlier critiques of the quality of services provided by LHWs [50], the integrated model specifically incorporated components of training, monitoring, and supportive supervision of LHWs to ensure sustained performance. A constructive feedback loop was generated for Lady Health Supervisors (LHS) to discuss their challenges and field issues and receive support, resources, and coaching accordingly [51–53]. This reflects in the improved indicators related to LHWs in the findings.

Despite the strength of this research, which utilized a large sample size for quasi-experimental design, the study has some limitations. The non-random allocation of units to the intervention and control arms may have affected internal validity, making it difficult to determine whether the observed effects were due to the intervention or other confounding factors. This was addressed by applying the DiD analysis to observe the impact of intervention by calculating interaction of pre post along with intervention and control comparison. As the negative sign showed in the indicators of ANC sought, and fully immunized variable that there was increase however not attributed to the intervention alone [54]. Further, difference in wealth quantiles between the intervention and control groups highlight existing disparities that may impact study findings. Another limitation is the difficulty in determining if difference in the control and intervention areas truly reflect the intervention's impact. Conducting a sensitivity analysis in future studies would help confirm the robustness of the main findings, even with different statistical methods. Furthermore, it is recommended to conduct a process evaluation. Although it was beyond the scope of this study, we acknowledge this as a limitation and an opportunity for future research.

This study encompassed the fact that the intervention was developed based on a systematic review and meta-analysis of FP interventions and modified reflecting local qualitative research conducted in the study population. This combined evidence shows that targeted interventions, such as strengthening existing services and integrating FP with MCH, are feasible in low-resource settings.

## Conclusion

The study concludes that integrating family planning (FP) with maternal, and child health (MCH) services proved impactful in increasing contraception uptake and minimizing missed opportunities. This also improves other MCH indicators such ANC, PNC and skilled birth attendance through evidence informed designed interventions and implementation. It underscores the necessity for cohesive efforts by the government and local stakeholders to design local, regional, and national policy frameworks pertaining to health and population planning for sustainable improvements.

## Supporting information

**S1 Table. Selection of control district using propensity score matching (PSM) from 23 rural districts of Sindh.**
(DOCX)

**S1 Data. XXX.**
(CSV)

**S1 Codebook. XXX.**
(XLS)

## Acknowledgments

We would like to acknowledge the Department of Health, Population Welfare Department, Government of Sindh, for their valuable collaboration in implementation of this project. We are also grateful for the study participants. Further, we also acknowledge our district-based staff for all their support in conducting project activities.

## Author contributions

**Conceptualization:** Zahid Ali Memon, Zulfiqar Bhutta, Hora Soltani.

**Data curation:** Wardah Ahmed, Muhammad Jawwad.

**Formal analysis:** Muhammad Jawwad.

**Funding acquisition:** Zulfiqar Bhutta.

**Methodology:** Wardah Ahmed, Shah Muhammad, Fatima Haider, Sophie Reale, Rachael Spencer, Hora Soltani.

**Project administration:** Zahid Ali Memon, Wardah Ahmed.

**Supervision:** Zahid Ali Memon, Shah Muhammad, Talib Hussain Lashari.

**Validation:** Talib Hussain Lashari, Muhammad Jawwad, Sophie Reale.

**Visualization:** Wardah Ahmed, Rachael Spencer.

**Writing – original draft:** Zahid Ali Memon.

**Writing – review & editing:** Zahid Ali Memon, Wardah Ahmed, Shah Muhammad, Fatima Haider, Talib Hussain Lashari, Sophie Reale, Rachael Spencer, Zulfiqar Bhutta, Hora Soltani.

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
